# Discovery of a long-ranged charge order with 1/4 Ge1-dimerization in an antiferromagnetic Kagome metal

Ziyuan Chen [1,7], Xueliang Wu [2,7], Shiming Zhou [1,7], Jiakang Zhang [1], Ruotong Yin [1], Yuanji Li [1], Mingzhe Li [1], Jiashuo Gong[1], Mingquan He [2], Yisheng Chai [2], Xiaoyuan Zhou [2], Yilin Wang [1,3,4], Aifeng Wang [2] ✉, Ya-Jun Yan [1,4] ✉ & Dong-Lai Feng [1,3,4,5,6] ✉

Exotic quantum states arise from the interplay of various degrees of freedom such as charge, spin, orbital, and lattice. Recently, a short-ranged charge order (CO) was discovered deep inside the antiferromagnetic phase of Kagome magnet FeGe, exhibiting close relationships with magnetism. Despite extensive investigations, the CO mechanism remains controversial, mainly because the short-ranged behavior hinders precise identification of CO superstructure. Here, combining multiple experimental techniques, we report the observation of a long-ranged CO in high-quality FeGe samples, which is accompanied with a first-order structural transition. With these high-quality samples, the distorted $2 \times 2 \times 2$ CO superstructure is characterized by a strong dimerization along the $c$-axis of 1/4 of Ge1-sites in $Fe_3Ge$ layers, and in response to that, the $2 \times 2$ in-plane charge modulations are induced. Moreover, we show that the previously reported short-ranged CO might be related to large occupational disorders at Ge1-site, which upsets the equilibrium of the CO state and the ideal $1 \times 1 \times 1$ structure with very close energies, inducing nanoscale coexistence of these two phases. Our study provides important clues for further understanding the CO properties in FeGe and helps to identify the CO mechanism.

A central theme of condensed matter physics is to search for novel phases of matter. Kagome lattice is composed of hexagons and triangles in a network of corner-shared triangles, known to host geometric frustration, nontrivial band topology, van Hove singularities (vHSs), and flat bands, hence it is a fertile platform to study the interplay of lattice, topology, magnetism and electron correlation[1,2]. Diverse phases have been discovered in Kagome materials, such as Mott insulator[2], quantum spin liquid[3], anomalous-Hall-effect system[4], unconventional superconductor[3,5,6], and topological semimetal and insulator[7–12]. Recently, charge density wave (CDW) states in Kagome metals have attracted extensive attention because of diverse CDW patterns[13–29], complex broken symmetries[17–20], and intertwining orders[13–16]. For instance, a $2 \times 2 \times 2$ CDW has been discovered in $AV_3Sb_5$ (A = K, Rb, Cs), with time-reversal symmetry breaking associated with a possible chiral flux phase, and was proposed to arise from Fermi surface nesting of vHSs[13–20]. Competing CDW instabilities have also been observed in

[1]Hefei National Research Center for Physical Sciences at the Microscale and Department of Physics, University of Science and Technology of China, Hefei, China. [2]Low temperature Physics Laboratory, College of Physics and Center of Quantum Materials and Devices, Chongqing University, Chongqing, China. [3]National Synchrotron Radiation Laboratory School of Nuclear Science and Technology, and New Cornerstone Science Laboratory, University of Science and Technology of China, Hefei, China. [4]Hefei National Laboratory, University of Science and Technology of China, Hefei, China. [5]Collaborative Innovation Center of Advanced Microstructures, Nanjing, China. [6]Shanghai Research Center for Quantum Sciences, Shanghai, China. [7]These authors contributed equally: Ziyuan Chen, Xueliang Wu, Shiming Zhou. ✉e-mail: afwang@cqu.edu.cn; yanyj87@ustc.edu.cn; dlfeng@ustc.edu.cn

ScV$_6$Sn$_6$, originating from strong electron-phonon couplings[21–29]. These CDW systems are weakly correlated, and magnetism does not play a noticeable role in their formation.

Recently, a charge order (CO) was discovered in a strongly correlated Kagome magnet FeGe[30–34], which is sometimes called CDW as it is in a metallic phase and possibly induced by Fermi surface instabilities[33]. It exhibits quite a few distinct features. Firstly, the CO is short-ranged with a correlation length of 2–4 nm, as revealed by neutron scattering and scanning tunneling microscopy (STM) studies[30,31,34]. Moreover, the CO can be easily disrupted by moderate bias (-1 V) during the STM measurements[34]. Secondly, the CO transition occurs deep inside the antiferromagnetic (AFM) phase, and the spin polarization is enhanced below the CO onset temperature ($T_{CO}$)[30]; while in comparison, the short-ranged COs in cuprates and nickelates appear above or at the magnetic ordering temperature[35–39]. Thirdly, an x-ray scattering study has observed a sharp superlattice peak at (0, 0, 2.5) below $T_{CO}$ with an obvious thermal hysteresis loop[32], indicating a structural transition along the c-axis. However, only a weak hump has been observed in the existing specific heat data at $T_{CO}$, suggesting a weak first-order transition. Several mechanisms have been proposed for the CO formation in FeGe, such as spin-phonon coupling[32], vHSs nesting[33], nontrivial topology effect[40], zero-point energy fluctuations[41], electron correlation[42], cooperation between electron correlation and electron-phonon couplings[43], the interplay of magnetism, structure and electron correlation[44], but it is still controversial due to the lack of decisive experimental evidences, such as the precise distorted CO superstructure, which is the first step in understanding the CO properties and mechanism. However, this is hindered, to a large extent, by the phase separation and small CO domains due to the short-ranged behavior[30,33,34].

Here, we report a long-ranged CO in high-quality FeGe single crystals, which show a sharp first-order structural phase transition at $T_{CO}$.

With the high-quality samples, we have successfully refined the distorted 2 × 2 × 2 CO superstructure by single crystal X-ray diffraction (SCXRD) and found that it is dominated by a strong dimerization along the c-axis of one-quarter of the Ge1-sites in the Kagome layers, consistent with the theoretical predictions via first-principle DFT calculations. Moreover, we show that the previously reported short-ranged CO in FeGe might be related to the existence of large occupational disorder of Ge1-site, which upsets the equilibrium of the CO state and the ideal 1 × 1 × 1 structure with very close energies, leading to the nanoscale coexistence of these two phases. Our study provides important clues for further understanding the CO properties in FeGe and helps to identify the CO mechanism.

## Results

### First-order phase transition of the long-ranged CO in FeGe

The black curve in Fig. 1a shows the temperature-dependent in-plane magnetic susceptibility of as-grown FeGe (sample #1). A weak drop appears around 100 K, which corresponds to the CO transition[30]. Accordingly, a weak hump appears in specific heat, but no obvious response is found in resistance (black curves in Fig. 1c, d). Consistent with the small CO domain sizes found by neutron scattering study[30], our STM study has observed strong phase separation and a short-ranged CO in sample #1 (see ref. 34 and Supplementary Fig. 1 of Supplementary Information (SI)). However, as shown in Fig. 1a, we find that the CO transition in FeGe is very sensitive to annealing treatment, the drop in susceptibility at $T_{CO}$ becomes much sharper as the annealing temperature decreases. It is the most significant for the samples annealed at 320 °C (sample #2), but almost absent when annealed at 560 °C. Figure 1b shows the thermal hysteresis scans of magnetic susceptibility for sample #2 around $T_{CO}$, where a hysteresis loop of ~0.75 K is obvious; meanwhile, a sharp and nearly divergent peak at $T_{CO}$ is observed in the specific heat data (Fig. 1c, blue curve), proving a

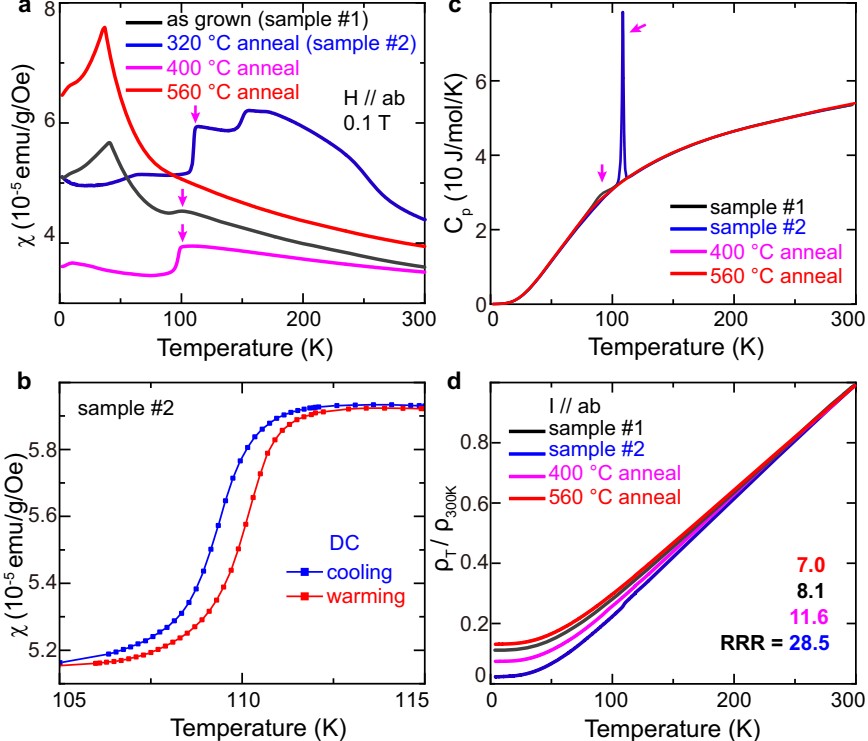

**Fig. 1 | Transport properties of various FeGe samples under different annealing treatments. a** Temperature dependent in-plane magnetic susceptibilities of as-grown FeGe (sample #1) and those after annealing at different temperatures as indicated. The FeGe sample annealed at 320 °C is labeled as sample #2. The CO transitions are indicated by the magenta arrows. **b** Hysteresis loop near $T_{CO}$ for sample #2, suggesting a first-order phase transition. **c** Temperature dependent specific heat data of different FeGe samples. The CO transitions are indicated by the magenta arrows. **d** Temperature dependent normalized in-plane resistance, $\rho_T/\rho_{300K}$, with their RRR values indicated.

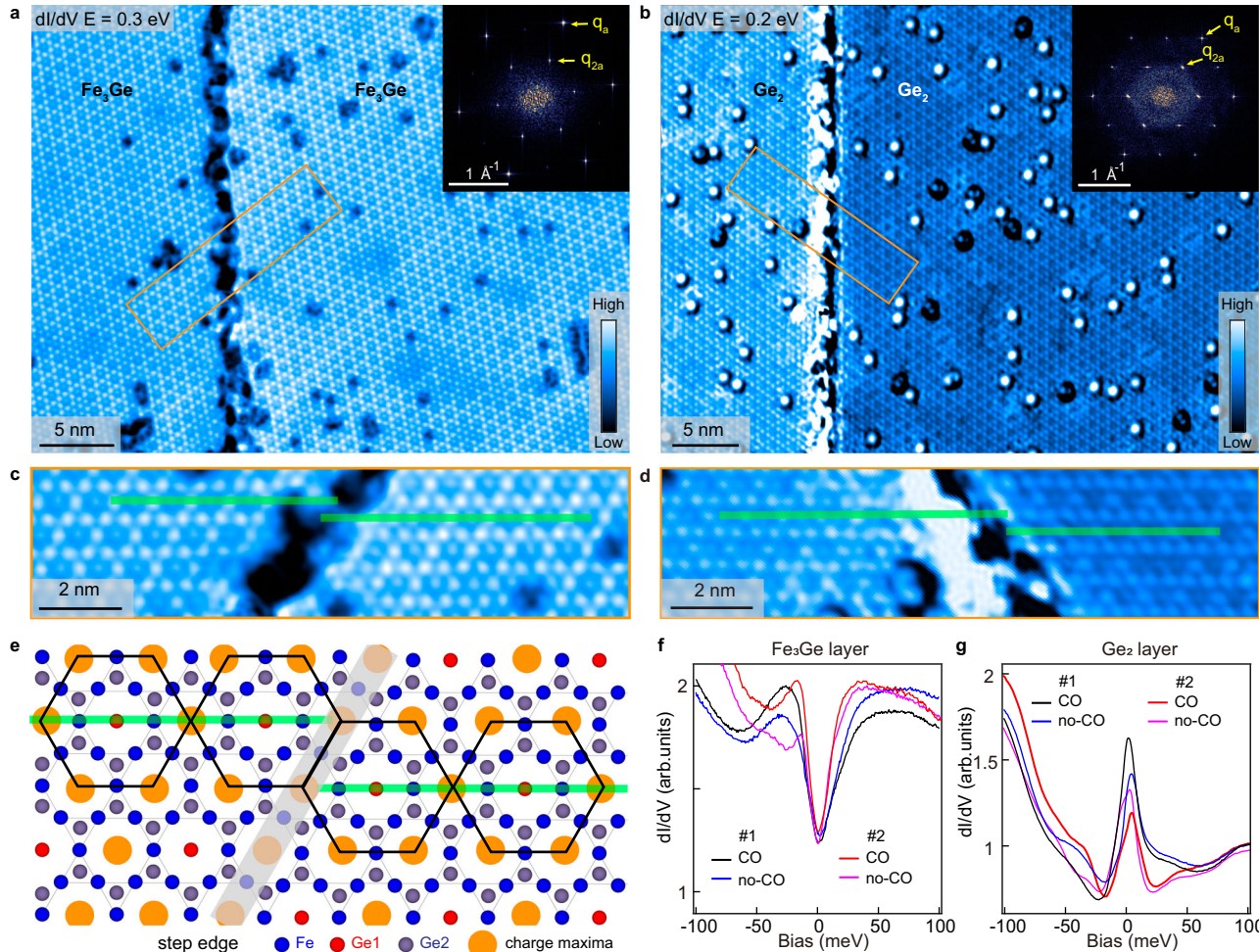

**Fig. 2 | Long-ranged CO revealed by STM. a**, **b** Typical *dI/dV* maps for the Fe₃Ge and Ge₂ layers of sample #2, both including a single-unit-cell-height step. Insets of (**a** and **b**) show their fast Fourier transform (FFT) images, with the Bragg spots of underlying lattice and CO spots labeled as $q_a$ and $q_{2a}$, respectively. **c**, **d** Enlarged images of the areas enclosed by the orange boxes in (**a** and **b**), respectively. **e** Illustration of the CO patterns near a single-unit-cell-height step. The orange circles represent the charge maxima, each black hexagon represents a minimal unit of the CO. Green lines in (**c**–**e**) track the chains with CO modulation, which shifts half unit-cell across the step edges. **f**, **g** Representative *dI/dV* spectra collected in the sample regions with or without CO in Fe₃Ge and Ge₂ layers of samples #1 and #2, respectively. Measurement conditions: (**a**) $V_b = 0.3$ V, $I_t = 300$ pA, $\Delta V = 20$ mV; (**b**) $V_b = 0.2$ V, $I_t = 300$ pA, $\Delta V = 30$ mV; (**f**, **g**) $V_b = 100$ mV, $I_t = 100$ pA, $\Delta V = 2$ mV.

typical first-order phase transition. Such a sharp transition behavior could be attributed to the improved sample quality. Figure 1d shows the normalized in-plane resistance curves of a series of annealed FeGe samples. The in-plane residual-resistance ratio (RRR) increases with decreased annealing temperature, and it is the largest for sample #2 (~28.5), indicating the improved sample quality. The enhancement of RRR value is usually related to the decrease of defect scattering, which will be discussed later. Besides, we notice that the magnetic behavior of FeGe is also strongly influenced by annealing treatment. A broad hump between 150 K and 260 K is observed in sample #2 (blue curve in Fig. 1a), which might be induced by the minor cubic B20-type FeGe byproduct probably formed from the low-temperature annealing process[45]. The magnetic behavior below $T_{CO}$ varies obviously for different samples, especially the canted AFM transition temperature, providing a good platform to study the interplay between magnetism and CO. However, as STM and SCXRD techniques used in this study are not sensitive to magnetism, neutron scattering, and other magnetic-sensitive techniques are required to reveal the mechanism.

Figure 2a, b show the typical *dI/dV* maps of the Fe₃Ge and Ge₂ layers of sample #2, demonstrating the CO distribution (see the topographic images and more *dI/dV* maps in Supplementary Figs. 2 and 3 of SI). Compared to sample #1, phase separation almost disappears in sample #2 and the 2 × 2 CO is long-ranged for both the Fe₃Ge and Ge₂ surfaces, as also reflected by the much stronger and sharper CO spots ($q_{2a}$) in the insets of Fig. 2a, b than those in Supplementary Fig. 1c. Moreover, the Fe₃Ge-Fe₃Ge and Ge₂-Ge₂ single-unit-cell-height step edges are observed, enabling direct detection of the CO periodicity along the *c*-axis. The close-up views of both steps are shown in Fig. 2c, d, and Fig. 2e illustrates the CO distribution nearby, which shifts half unit-cell across the step edges, revealing a relative π-phase shift. These observations establish a three-dimensional 2 × 2 × 2 CO state. Moreover, we measure the *dI/dV* spectra in sample regions with or without CO for both the Fe₃Ge and Ge₂ layers of samples #1 and #2, respectively, as displayed in Fig. 2f, g. A partially opened gap-like feature is observed in all the Fe₃Ge regions with or without CO, but it is absent in the Ge₂ layer, which instead shows a sharp peak around $E_F$. This suggests that the gap-like feature in Fe₃Ge layer may not be associated with the CO gap. This is consistent with recent Raman scattering and angular resolved photoelectron spectroscopy studies, where a CO gap was not observed[46–48]. Overall, by improving the sample quality, we have shown that a long-ranged CO does exist in FeGe, and the CO phase transition is of the first order.

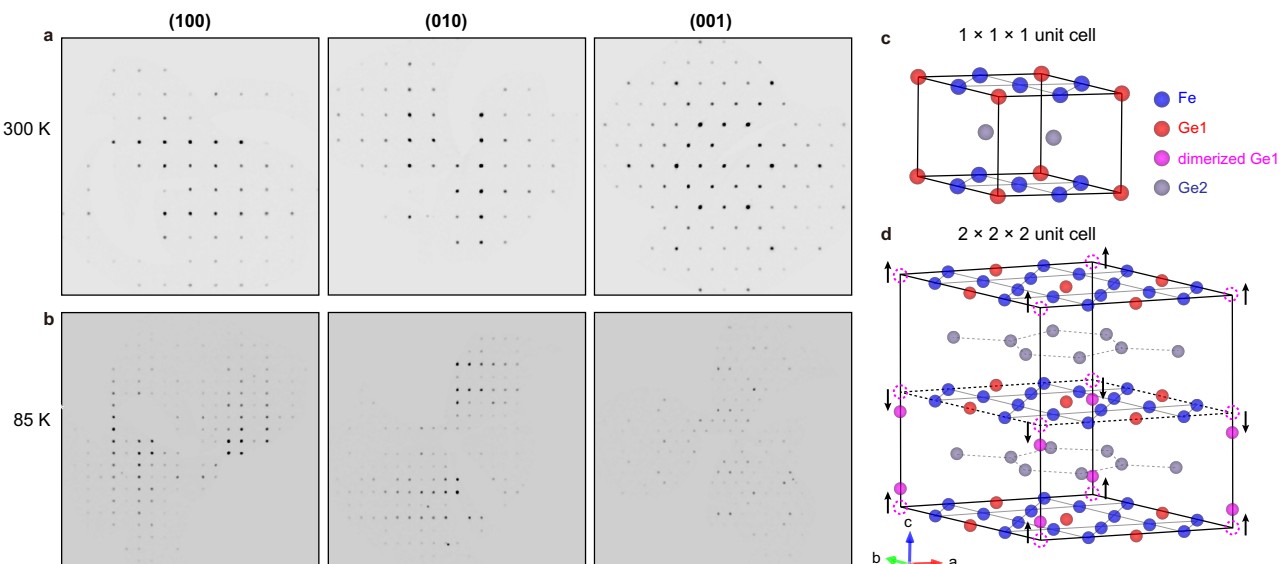

**Fig. 3 | 2 × 2 × 2 CO superstructure revealed by SCXRD experiments on sample #2. a, b** Diffraction patterns along the *a*-, *b*-, and *c*-axes of sample #2 at 300 K and 85 K, respectively. New spots appear at 85 K and signal a structural modulation with a wave vector of (−0.5, 0.5, 0.5). **c, d** Refined crystal structures at 300 K and 85 K, respectively. 1/4 of Ge1 atoms in the Kagome layers show a large dimerization (-0.7 Å) along the *c*-axis and the other atoms undergo small distortions, which doubles the lattice in all three lattice directions. The dashed magenta circles indicate the positions of Ge1 atoms before dimerization, the black arrows show the directions of Ge1 movement.

## Structural deformation in the CO phase

A key step in understanding the CO is to determine the atomic positions of the distorted superstructure. We performed SCXRD measurements on both sample #1 and sample #2 at 300 K and 85 K, respectively, and the corresponding crystal structures were solved and refined. Considering that the change of physical properties of FeGe by annealing is reversible[45,49] and the defect density observed by STM study is less than 2% as discussed in Supplementary Figs. 4 and 5 of SI, we first adopt the full FeGe stoichiometry for simplicity. Reliable high-quality refinement results were obtained, as demonstrated by the values of final R indexes ($R_1$ and $wR_2$ in Supplementary Tables 1 and 4 of SI), and the detailed results are shown in Supplementary Figs. 6–8 of SI. We also consider the effect of defects, which improves the refinements moderately but does not alter the key structural features, as shown in Supplementary Fig. 9 of SI and will be discussed in detail later.

Although sample #1 exhibits a weak CO anomaly in susceptibility and specific heat (Fig. 1a, c), our SCXRD experiment could not resolve obvious superstructure diffraction spots at 85 K (Supplementary Fig. 6 of SI) and find the similar dominated 1 × 1 × 1 structure (Fig. 3c) at both 300 K and 85 K (Supplementary Tables 1–3 of SI). This discrepancy could be attributed to the strong nanoscale phase separation of the 2 × 2 × 2 CO phase and the dominant 1 × 1 × 1 phase in sample #1, which makes the superstructure spots too weak to detect. Likewise, the lattice superstructure could not be retrieved in the previous x-ray diffraction measurement on FeGe that found a superlattice peak at (0, 0, 2.5) (ref. 32). On the other hand, new diffraction spots are observed below $T_{CO}$ for sample #2. Figure 3a, b present the diffraction patterns along the *a*-, *b*-, and *c*-axes of sample #2 at 300 K and 85 K, respectively. At 85 K, the additional diffraction peaks signal a structural modulation with a wave vector of (−0.5, 0.5, 0.5), consistent with the observed commensurate 2 × 2 × 2 CO superstructure.

The refined crystal structures of sample #2 at 300 K and 85 K are sketched in Fig. 3c, d. The room temperature structure possesses a space group of P6/*mmm* with *a* = *b* = 4.995 Å and *c* = 4.053 Å, consistent with the previous report[30]. Moreover, we find that the refined 2 × 2 × 2 CO superstructure (Fig. 3d) is mainly dominated by a large dimerization of 1/4 of the Ge1-sites in the Kagome layers along the *c*-axis.

We note that the same Ge1-dimerization superstructure (space group P6/*mmm*) has been theoretically predicted via first-principle DFT calculations[32,41,44]. The refined dimerized Ge1-sites are 0.7 Å away from the Kagome plane. Such a substantial deformation is yet very close to the theoretically predicted value of 0.65 Å. The theory also predicted a Kekulé-type distortion of Ge2-sites in the honeycomb layers and distortions of Fe-sites along the *c*-axis, with opposite phases between adjacent Kagome (honeycomb) layers, but even the largest distortions are below 0.05 Å. Such distortions are basically reproduced in our SCXRD results as well, but the refined distorted structure has a slightly lower symmetry (space group P-6*m*2) than P6/*mmm* (see Supplementary Tables 4–6 and Supplementary Fig. 8 of SI for more details). Using this refined structure as an initial guess, we performed structural relaxation by DFT calculations and it eventually converged to the theoretically predicted one with a higher symmetry (P6/*mmm*). Therefore, this deviation might be related to the limitation of imperfections in the sample or experimental precision since those distortions are very small, which needs further investigation; however, the main feature, namely the large Ge1-dimerization, is in good agreement with the theory, where such a dimerization is crucial for the CO formation in FeGe.

## STM evidence of structural distortion

Like the easily disrupted short-ranged CO in sample #1 (see ref. 34), the long-ranged 2 × 2 × 2 CO in sample #2 can also be disrupted by STM scanning or hovering the STM tip atop with a mildly higher bias of $|V_b| \geq 0.9$ V, and a sizable nearby region transforms into the 1 × 1 × 1 phase, as shown in Fig. 4a–d, and see more in Supplementary Figs. 10 and 11 of SI). The exact disruption mechanism is unclear now. We suspect that the larger electric field or electron/hole injections induced by applying a high $|V_b|$ might upset the balance of the 2 × 2 × 2 CO and the 1 × 1 × 1 phase with nearly degenerate energies, which causes the phase transition[34]. Based on our SCXRD data, the doubled lattice parameters of the 1 × 1 × 1 phase are slightly larger than those of the 2 × 2 × 2 CO phase by 1.7 pm along the *a*-axis and 1.3 pm along the *c*-axis. Because STM has a high spatial resolution perpendicular to the surface, such a structural difference in different phases could be retrieved. Figure 4e–g shows the typical topographic images of the

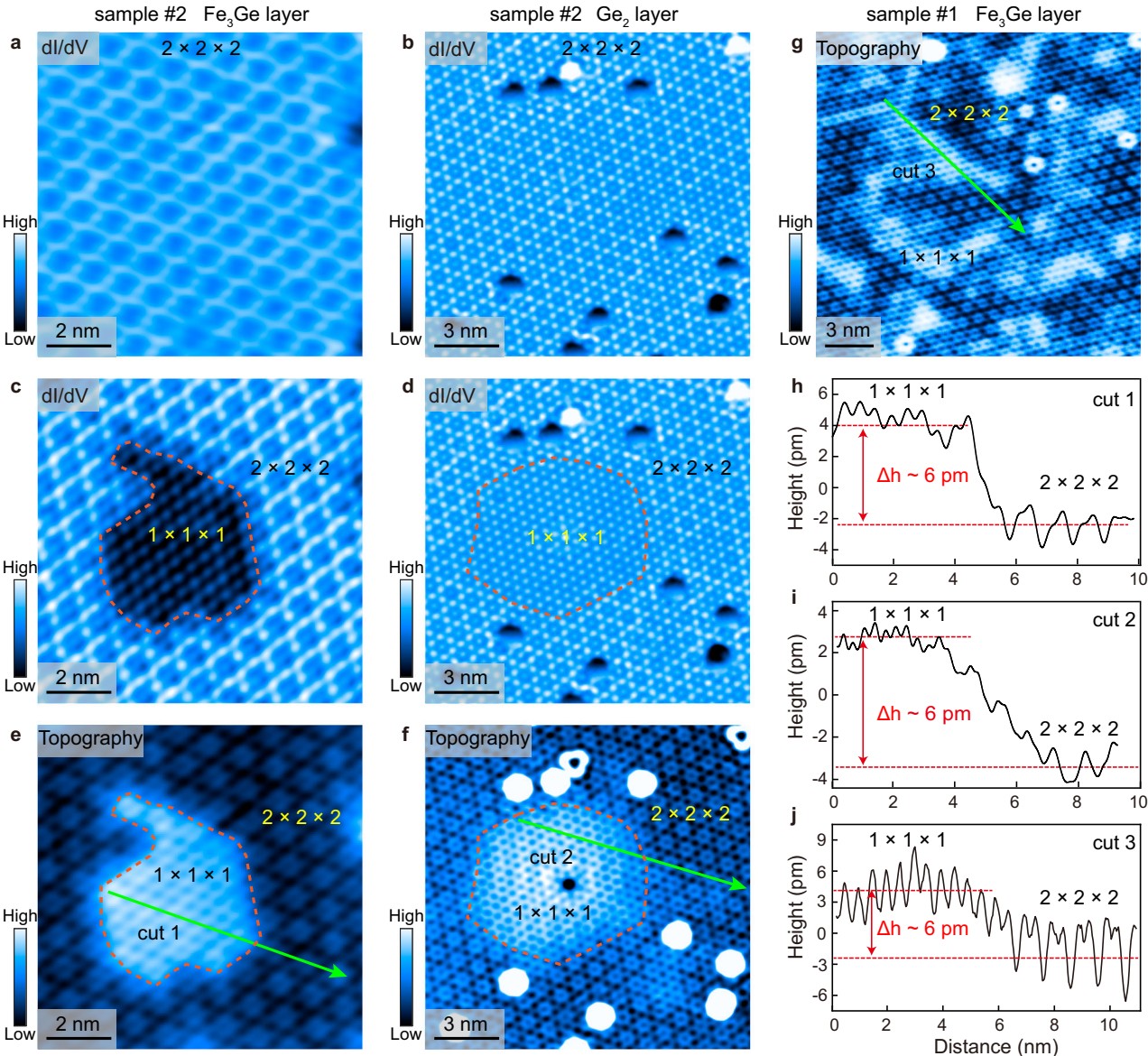

**Fig. 4 | Phase transformation from the 2 × 2 × 2 CO to the 1 × 1 × 1 phase.**
**a**, **b** Typical CO modulations in the Fe₃Ge and Ge₂ layers of sample #2.
**c**, **d** Disrupted CO distributions in the same regions of (**a**, **b**). CO modulation disappears in the areas enclosed by the orange dashed curves. **e**, **f** Corresponding topographic images of (**c**, **d**). **g** Typical topographic image of the Fe₃Ge layer in sample #1, exhibiting strong phase separation. **h–j** Lattice line profiles taken along cuts 1–3. The height difference of the two phases, $\Delta h$, is marked out. Measurement conditions: (**a–d**) $V_b = 0.3$ V, $I_t = 300$ pA, $\Delta V = 30$ mV; (**e**, **f**) $V_b = 0.3$ V, $I_t = 300$ pA; (**g**) $V_b = 50$ mV, $I_t = 20$ pA.

phase separation regions in both samples #1 and #2 and more datasets collected under a wide energy range are displayed in Supplementary Fig. 11 of SI. Under all these energies, the 2 × 2 × 2 CO is higher than the 1 × 1 × 1 phase in local density of states but is lower in STM topography. This suggests that the STM topography truly reflects the relative height difference between these two phases, and the *c*-axis lattice parameter of the 1 × 1 × 1 phase is slightly larger than that of the 2 × 2 × 2 CO. Moreover, a consistent height difference of $\Delta h$ ~ 6–9 pm is revealed for both the Fe₃Ge and Ge₂ layers of the two samples by extracting the lattice line profiles, as shown in Fig. 4h–j and Supplementary Fig. 11 of SI. The larger height difference implies that several CO units along the *c*-axis are transformed into the 1 × 1 × 1 phase, and surface relaxations may also contribute.

## Defect effects on CO
The long-ranged CO in sample #2 indicates that the Ge1-dimerization is established over a large region after annealing, while it is short-ranged

in sample #1. As discussed above, the RRR value of sample #2 is increased, thus the density of defects should be reduced in sample #2 compared with sample #1. Besides, considering that the change of physical properties of FeGe by annealing is reversible[45,49], the FeGe stoichiometry will not change much during these processes. It is more likely that annealing treatments change mainly the occupational disorders of atomic sites. Next, we analyze the defect distribution in samples #1 and #2 by STM and SCXRD, to try to reveal their effect on CO.

Figure 5a–d shows the typical topographic images of the Fe₃Ge and Ge₂ layers for both samples #1 and #2, demonstrating the defect distribution. The native defects are indicated by the black or yellow arrows, while the bright spots are residual atoms after cleavage, which have little influence on the CO distribution. By measuring the detailed topographic images under a wide energy range, we are able to count and classify these defects, please see Supplementary Figs. 4 and 5 of SI for more details. The Ge1-site defects are most often observed, and

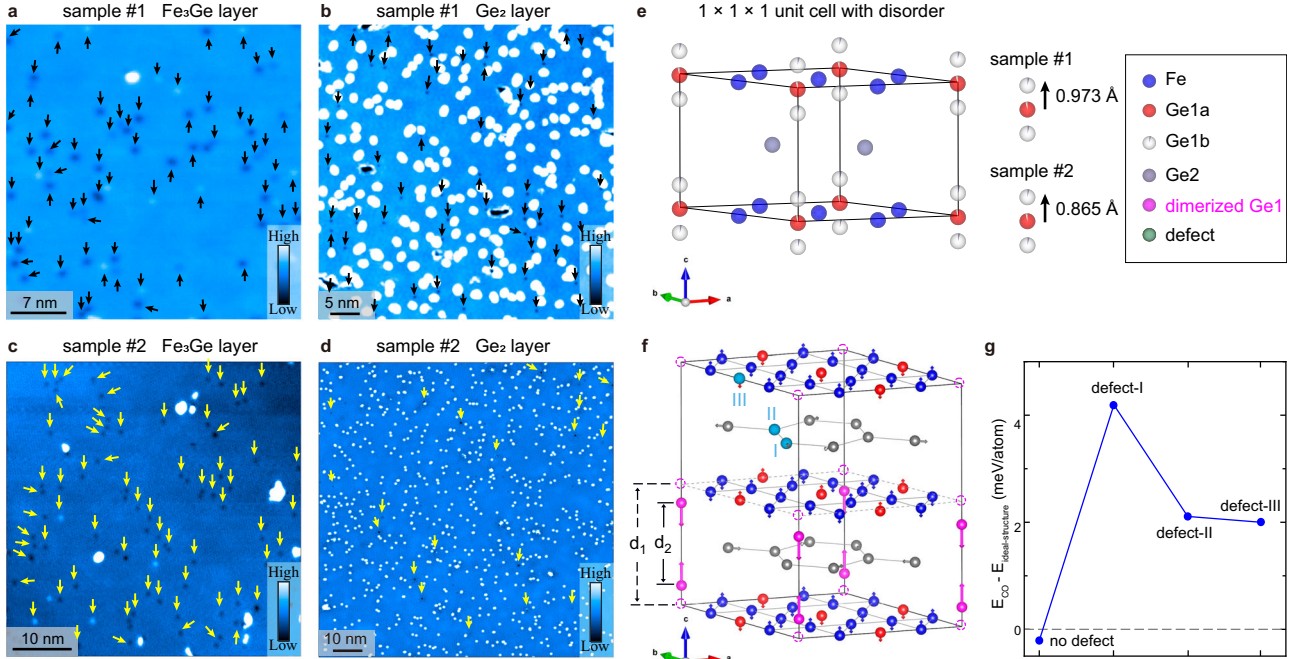

**Fig. 5 | Defect distribution and effect on CO. a–d** Typical topographic images of the Fe₃Ge and Ge₂ layers of samples #1 and #2, where the native defects are indicated by the black or yellow arrows. The bright spots are residual atoms after cleavage, which have little influence on CO distribution. **e** Corrected crystal structure of FeGe at 300 K, considering the occupational disorder of Ge1-site. Partial Ge1 atoms (labeled as Ge1b) move in the $c$-axis away from the Kagome plane, their proportion to the total Ge1-site and the averaged distance from the Kagome plane are ~3% and 0.973 Å for sample #1 and ~2% and 0.865 Å for sample #2, respectively, as sketched in the right panel of (**e**). **f** Theoretical predicted 2 × 2 × 2 CO superstructure with large 1/4 Ge1-dimerization, as indicated by the magenta longer arrows. The dimerization strength is defined as the difference of Ge1-Ge1 bond lengths before and after dimerization, $\Delta d = d_1 - d_2$. $\Delta d = 0$ represents the ideal structure without Ge1-dimerization, $\Delta d = 1.3$ Å corresponds to the actual value of the CO state. The shorter arrows indicate the small distortions of other sites associated with the in-plane 2 × 2 charge modulations. Possible Ge2-defects (I, II) and Ge1-defect (III) are shown by the sky-blue color. **g** The DFT calculated energy difference between the CO state and the ideal 1 × 1 × 1 structure for $\Delta d = 1.3$ Å, without defects and with one defect at position I, II, or III, respectively. Measurement conditions: (**a**, **b**) $V_b = 1.0$ V, $I_t = 10$ pA; (**c**) $V_b = 1.0$ V, $I_t = 300$ pA; (**d**) $V_b = 40$ mV, $I_t = 300$ pA.

their total density is reduced by approximately a third to a half in sample #2, from ~1.7% of the atomic proportion of Ge1-site in sample #1 to ~0.8% in sample #2 (Supplementary Fig. 5 of SI). Fe-site defect is less common, and the density of Ge2-site defect is very low compared to that of Ge1-site defect. The estimated density of Ge2-site defects is less than 0.01% of the atomic proportion of Ge2-site and is almost unchanged in both samples, which should have little influence on CO.

Besides, when scrutinizing the SCXRD data collected at 300 K, we find that there is moderate residual electron density around Ge1b-site (corresponding to the Ge1 atoms that move along the $c$-axis away from the Kagome plane, the unchanged Ge1 atom is labeled as Ge1a-site, as illustrated in Fig. 5e), suggesting obvious occupational disorders at Ge1-site, as also discussed in Supplementary Fig. 9 of SI. By adding such occupational disorders at Ge1-site, the refinements are moderately improved (see Supplementary Table 7 of SI). The obtained proportion of Ge1b-site to the total Ge1-site and their averaged distance from the Kagome plane are ~3% and ~0.973 Å for sample #1 and ~2% and ~0.865 Å for sample #2, respectively, as also illustrated in the right panel of Fig. 5e. Such occupational disorders of Ge1-site have been reported by other SCXRD measurements[45]. Other types of defects, such as Ge2-site defects, do not show up in SCXRD data and the artificial introduction of a small amount of them has little effect on the refinements. These results qualitatively agree with our STM observations—the main defects in FeGe are Ge1-site defects, which are obviously reduced in sample #2. The decrease of defects in sample #2 is considered responsible for the increased RRR value.

Finally, we check whether the different types of defects may alter the relative energy between the CO and the 1 × 1 × 1 phases. We simulate the effect of several typical defects such as Ge2-vacancies and Ge1-site defects by DFT calculations in a 2 × 2 × 2 superstructure (see Fig. 5f). Figure 5g shows the calculated energy difference between the CO state and the ideal 1 × 1 × 1 structure for the Ge1-dimerization strength $\Delta d = 1.3$ Å. Without Ge1- or Ge2-site defects, the CO is the ground state; but its energy is very close to the original 1 × 1 × 1 phase, which may explain the easy disruption of the CO state discussed in Fig. 4. In the presence of one Ge1- or Ge2-site defect (type I, II or III in Fig. 5f), the CO possesses a much higher energy than that of the ideal 1 × 1 × 1 structure, indicating that such defects prevent the large dimerization of Ge1-sites and the formation of CO. Overall, the results of our STM, SCXRD and DFT calculations are qualitatively consistent, which suggest that the dense defects in sample #1, especially the large Ge1b-site disorder, prevent the formation of the large domains with 1/4 Ge1-dimerization, resulting in the nanoscale phase separation and a short-ranged CO.

## Discussion
In summary, we report a long-ranged CO in high-quality FeGe single crystals, accompanied with a sharp first-order structural phase transition. The distorted 2 × 2 × 2 CO superstructure is precisely determined, which is dominated by a strong dimerization along the $c$-axis of one-quarter of the Ge1-sites in the Kagome layers, and in response to that, the 2 × 2 in-plane charge modulations are induced. This CO superstructure is consistent with the theoretical predictions via first-principle DFT calculations[32,41,44]. Based on these theoretical analyses, the large dimerization of Ge1-sites along the $c$-axis will enhance the electronic correlations of Fe-3$d$ orbitals and, as a result, it enhances the spin polarization of the AFM state, which reproduces well the results reported by neutron scattering study[30]. Furthermore, we show that the previously reported short-ranged CO in FeGe might be related to the existence of a large occupational disorder of Ge1-site, which upsets the

equilibrium of the CO state and the ideal $1 \times 1 \times 1$ structure with very close energies. The short-ranged behavior of the CO in FeGe is extrinsic, different from those observed in cuprates and nickelates. Our study provides crucial clues for further understanding the CO properties in FeGe and helps to identify the CO mechanism. Given the diversity of FeGe-type and derived $HfFe_6Ge_6$-type compounds, there are many opportunities to tune the CO state to uncover the influence factors and to gain a deeper understanding of the interplay between lattice, magnetism and electronic correlations.

## Methods

### Synthesis and anneal of FeGe crystals
Single crystals of B35-type FeGe were synthesized via the chemical vapor transport (CVT) method. Iron powders (99.99%) and germanium powders (99.999%) were weighed and mixed in a stoichiometric ratio of 1:1 with additional iodine as transport agents. These starting materials were sealed into a silicon quartz tube under a high vacuum and placed horizontally into a two-zone furnace. The source and sink temperatures for the growth were set at 600 °C and 550 °C, respectively, and kept for 2 weeks. After cooling naturally to room temperature by switching off the furnace, the as-grown shiny FeGe single crystals with a typical dimension of $2 \times 2 \times 2$ mm³ can be obtained in the middle of the quartz tube. For the post-growth annealing protocols, we selected 560, 400, and 320 °C as the annealing temperatures, and the annealing time is kept the same for 48 h. The obtained as-grown single crystals were divided into several portions and separately sealed into quartz tubes under a high vacuum. After being kept at the target temperature for 48 h, the tube was immediately taken out of the furnace and quenched into water. It is worth mentioning that the physical properties of the annealed crystals are determined merely by the final annealing temperature, regardless of the behavior of the starting crystals, as-grown or annealed crystals, used for annealing.

### Sample characterizations
Magnetic susceptibility was measured using a direct current scan mode for Fig. 1b, and a vibrating sample magnetometer option for Fig. 1a in the Quantum Design Magnetic Property Measurement System. Specific heat and resistivity measurements were conducted in a Quantum Design DynaCool Physical Properties Measurement System (PPMS-9T). Resistivity was measured in a standard four-probe configuration.

### SCXRD measurements
SCXRD measurements were performed at 300 K and 85 K with a Rigaku SuperNova diffractometer using Mo Kα radiation and an Oxford Cryosystem cooler. The diffraction data were collected and reduced with Rigaku Oxford Diffraction CrysAlisPro software[50]. The crystal structures were solved and refined with Olex2 software[51] and Shelx program[52].

### STM measurements
FeGe crystals were mechanically cleaved at 80 K in ultrahigh vacuum with a base pressure better than $1 \times 10^{-10}$ mbar and immediately transferred into a UNISOKU cryogenic STM at $T = 4.2$ K. Pt-Ir tips were used after being treated on a clean Au (111) substrate. The topographic images were obtained by a constant-current mode. The $dI/dV$ spectra were collected by a standard lock-in technique with a modulation frequency of 973 Hz and a typical modulation amplitude $\Delta V$ of 2-30 mV at 4.2 K.

### DFT calculations
DFT calculations were performed using the Vienna ab initio simulation package (VASP)[53]. The exchange-correlation potential is treated within the generalized gradient approximation of the Perdew–Burke–Ernzerhof variety[54]. The internal atomic positions of the charge-dimerized $2 \times 2 \times 2$ superstructure are relaxed until the force is less than 0.001 eV/Å for each atom. Integration for the Brillouin zone is done using a Γ-centered $8 \times 8 \times 10$ $k$-point grids for the $2 \times 2 \times 2$ supercell and the cutoff energy for plane-wave-basis is set to be 500 eV.

## Note added to proof
During the review process of this manuscript, we noticed another work by Shi et al.[45] reporting the similar annealing effect on achieving the long-ranged CO in FeGe. And they also obtained a similar CO superstructure and revealed the occupational disorder of Ge1-site by SCXRD measurements.

## Data availability
All the data supporting the findings of this study are provided within the article and its Supplementary Information files. All the raw data generated in this study are available from the corresponding author upon request.

## Code availability
All the data analysis codes related to this study are available from the corresponding author upon request.

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

## Acknowledgements

We thank Prof. Tong Zhang and Prof. Yuan Li for helpful discussions. This work is supported by the National Natural Science Foundation of China (Grants Nos. 12074363 (Y.J.Y.), 11790312 (D.L.F), 12174365 (Y.L.W.), 12004056 (A.F.W.), 11888101 (D.L.F), 11774060 (Y.J.Y)), the Innovation Program for Quantum Science and Technology (Grant No. 2021ZD0302803 (D.L.F.)), the Fundamental Research Funds for the Central Universities of China (Grant No. 2022CDJXY-002 (A.F.W.), WK9990000103 (Y.J.Y.)), the New Cornerstone Science Foundation (D.L.F.), and Chongqing Research Program of Basic Research and Frontier Technology, China (Grants No. cstc2021jcyj-msxmX0661 (A.F.W.)).

## Author contributions

Growth, annealing treatment, and transport measurements of FeGe single crystals were performed by X. W. under the guidance of A. W.; SCXRD measurements and crystal structure analyses were performed by S. Z.; STM measurements were performed by Z. C., J. Z. under the guidance of Y. Y.; DFT simulations were performed by Y. W.; The data analysis was performed by Z. C., Y. Y., Y. W., D. F., J. Z., Y. Li, R. Y., M. L., J. G., Y. C., M. H., and X. Z.; A. W., Y. W., Y. Y. and D. F. coordinated the whole work and wrote the manuscript. All authors have discussed the results and the interpretation.

## Competing interests

The authors declare no competing interests.
