## [Transparent Peer Review file · Nature Communications]

Discovery of a long-ranged charge order with 1/4 Ge1-dimerization in an antiferromagnetic Kagome metal

Corresponding Author: Professor Ya-Jun Yan

Version 0:

Reviewer comments:

Reviewer #1

(Remarks to the Author)

Kagome materials have attracted significant attentions as a novel platform for studying the intertwined states including charge, spin and superconducting pairing. Ziyuan Chen et al. report the first observation of long-ranged 2x2x2 charge order characterized by a strong Ge-1 dimerization in annealed FeGe samples. The short range of charge order in FeGe has hindered the deep understanding of its nature. The authors improve the sample quality through annealing at different temperature. By combining transport, STM, SCXRD and DFT calculation, the authors reveal that the presence of Ge-2 defects leading to nanoscale phase separation of a 2x2x2 superstructure and an ideal 1x1x1 structure. This extrinsic origin of short-ranged charge order differs from that in cuprates and nickelates. The data presented here are of high quality and contribute to further understanding the charge order in FeGe. The manuscript is well written, I'm, however, not persuaded to recommend the current version to publish in Nature communications for the following concerns:

1. The authors utilize the height differences between the 2x2x2 charge order phase and 1x1x1 phase to reveal the structural distortion in the c-axis. However, as we know that, the topography in STM represents an integral of LDOS between the bias and Fermi level. The heights or height differences in STM topography may not represent a "real" height because it could be induced by the different LDOS in various components.

2. It would be beneficial if the authors could provide some dI/dV curves of sample #2 and sample #1. Is there any feature of CDW gap in the long-ranged charge order phase?

3. While the long-ranged charge order in sample #2 appears inert to the defect density, its absence in the as-grown sample #1 with dense defect density need more detailed discussion. The authors should elaborate on how defect density affects long-ranged charge order.

4. As shown in Fig. 1a, the magnetic susceptibilities of different annealed samples show very different behaviors, such as the transition temperature of canted AFM changes a lot with the different annealed temperature, and an additional transition around 150 K in sample #2. Further discussion on these significant changes is encouraged.

5. The disappearance of canted AFM in sample #2 coincides with the observation of the long-ranged charge order. The authors should explore if the canted AFM have any influence on the disappearance of long-ranged charge order.

Reviewer #2

(Remarks to the Author)

This paper reports careful transport, annealing, STM, X-ray scattering experiments to study the annealing effect on charge density wave (CDW) of FeGe, and establishment of long range CDW order in FeGe and lattice distortion induced by CDW. Over the past several years, CDW has been observed in several kagome lattice materials, and CDW in FeGe is particularly interesting because it appears at temperatures well below magnetic ordering temperature T_N . In the original paper, CDW in FeGe was reported to be short range order (refs. 30-34). The authors used different annealing temperature to show that low-temperature annealing can drive the system into long range CDW order, and solved lattice structure distortion induced by CDW order. Overall, the paper is well written and I think worthy of publication in Nature Communications. I do have some questions that authors should address before the paper is accepted for publication.

1. The authors use STM to show that defect density is much lower in annealed sample with full CDW order. However, it is not clear at all what defects the authors are talking about? Are these defects Fe or Ge vacancies or excess Fe/Ge in the

sample? This needs to be resolved to convince the reader about authors' story.

2. In the refinement obtained by authors, they claimed that the main effect of CDW order is c-axis dimerization, consistent with theory and previous X-ray work (Ref. 32). However, the refinements are done with full FeGe stoichiometry, how is this consistent with STM claim that annealed sample with full CDW would have less defects? The X-ray work seems to indicate that the sample stoichiometry is unchanged across CDW. Why would some samples show full CDW while other annealed samples show no CDW? How do one understand this from STM or X-ray scattering point of view? They seem to be inconsistent from the current story of the authors.

3. I am particularly interested in knowing what are the differences in crystal structures of FeGe for samples with short and long-range CDW order. From STM measurements, the authors suggest as grown samples have considerable defects in Ge2 sites, how would that affect the refinements and how to understand the effects of these defects to CDW order?

4. The DFT calculations suggest defect 2 has lower energies and that prevent the formation of $\frac{1}{4}$ Ge1 dimerization and CDW order. While this is a nice picture, I don't understand how this can be consistent with Fig. 4 where STM current can perturb the CDW order? I would assume STM tip current would not be able to affect defects in FeGe. There should be a detailed discuss to make sure the story is self-consistent between different probes.

5. Are there signatures of Ge2 defect concentration from X-ray data? Are they consistent with STM work?

6. I noticed that a recent preprint claiming a lattice distortion in FeGe (<https://arxiv.org/abs/2309.14314>). Is it consistent with authors X-ray work?

7. I also noticed a competing paper (<https://arxiv.org/abs/2308.09034>) posted on arxiv. The authors should make some comments near the end about their work and the competing work, so a general audience would know what is going on in the field.

Overall, the experimental part of the paper is nice and has considerable new information. However, it is not clear to me how different probes are self-consistent. The paper is not suitable for publication as written. A revised version of the paper might be suitable for publication.

Reviewer #3

(Remarks to the Author)

The authors present a study that reveals a way to tune the recently discovered charge order in the magnetic kagome system FeGe into a long range order by thermal annealing. Understanding emergent orders, especially charge order, in kagome lattices is an important current topic in condensed matter physics. The CO in FeGe is especially interesting as it is manifested in a moderately correlated system. The authors present a systematic comparison of an as-grown FeGe and a thermally annealed FeGe. First, they show that the annealing process reduces the density of defects via an STM study, which is responsible for the tuning of the CO. Second, they show that the CO exhibits a Ge dimerization as previously proposed. They do this by purposely destroying a region of CO using the STM tip, and then examining the height differences. Third, as defects exist in both the Fe3Ge layer and the Ge2 layer, they also explored which defects are more important for the CO stability, and made the observation that the defects in the Ge2 layer is more important, and support this with DFT calculations. Overall, this is a very important and systematic study. It provides key information to support the importance of Ge dimerization to the CO in FeGe, and also provides insights into why thermal annealing can be used to tune CO in this system. I find the analyses thorough and solid. I would recommend publication of this work. I only have the following minor questions below.

What are the native defect states shown in Fig. 1 e-f? And what are the residual atoms as the bright spots? Do the authors have any idea if they are Fe or Ge? It is not clear to me why cleaving would leave such residual atoms behind. Can STS taken on these defects provide any insights into what these two types of defects are?

From the susceptibility plot in Fig. 1a, it appears that the annealing conditions also strongly affect the feature well below 100K. There is a big peak around ~40K (?) in the as-grown crystal, which is basically preserved in the 560C sample, but gone in the 320C and strongly suppressed in the 400C sample. Do the authors have any understanding of what this is and how it is affected as well by the annealing process?

Author Rebuttal letter:

Reply to the reviewers:

We thank all reviewers for their time and insightful comments on our manuscript. Our point-by-point responses are in blue text below and the original comments are in italic. The corresponding revisions in manuscript are highlighted with yellow background.

Replies to the first reviewer's report:

Comment 1: Kagome materials have attracted significant attentions as a novel platform for studying the intertwined states including charge, spin and superconducting pairing. Ziyuan Chen et al. report the first observation of long-ranged $2\sqrt{2}\times 2\sqrt{2}$ charge order characterized by a strong Ge-1 dimerization in annealed FeGe samples. The short range of charge order in FeGe has hindered the deep understanding of its nature. The authors improve the sample quality through annealing at different temperature. By combining

transport, STM, SCXRD and DFT calculation, the authors reveal that the presence of Ge-2 defects leading to nanoscale phase separation of a $2\sqrt{2}\sqrt{2}$ superstructure and an ideal $1\sqrt{2}\sqrt{2}$ structure. This extrinsic origin of short-ranged charge order differs from that in cuprates and nickelates. The data presented here are of high quality and contribute to further understanding the charge order in FeGe. The manuscript is well written, I am, however, not persuaded to recommend the current version to publish in Nature communications for the following concerns.

Reply: We thank the reviewer for the comprehensive summary of the key results of our manuscript and giving high regard of our work. The reviewer puts forward some insightful comments which help us improve our manuscript substantially. Below we respond to the reviewer's comments point-by-point, and accordingly, we added additional datasets and improved the analysis/discussion. We believe our reply and the new version of our manuscript can eliminate the reviewer's doubts.

Comment 2: The authors utilize the height differences between the $2\sqrt{2}\sqrt{2}$ charge order phase and $1\sqrt{2}\sqrt{2}$ phase to reveal the structural distortion in the c-axis. However, as we know that, the topography in STM represents an integral of LDOS between the bias and Fermi level. The heights or height differences in STM topography may not represent a "real" height because it could be induced by the different LDOS in various components.

Reply: This is a good and professional question. Yes, the STM topography may be the superposition of the real topography of sample surface and an integral of LDOS between the bias and EF. In our study, we have considered this point carefully. We have measured both the STM topography and LDOS map of the $2\sqrt{2}\sqrt{2}$ CO and the $1\sqrt{2}\sqrt{2}$ phases over a wide energy range, as shown in Fig. 4 of the revised main text and Fig. S11 of the revised SM file. We find that in LDOS maps under all the measured energies, the LDOS of the $2\sqrt{2}\sqrt{2}$ CO is higher than that of the $1\sqrt{2}\sqrt{2}$ phase, indicating that the integral of LDOS of the $2\sqrt{2}\sqrt{2}$ CO should be higher than that of the $1\sqrt{2}\sqrt{2}$ phase. If the integral of LDOS contributes dominantly to the STM topography, we expect to see a higher height in the $2\sqrt{2}\sqrt{2}$ CO phase compared to the $1\sqrt{2}\sqrt{2}$ phase. However, what we see in STM topography is opposite --- the $2\sqrt{2}\sqrt{2}$ CO phase is lower than the $1\sqrt{2}\sqrt{2}$ phase over a wide energy range. Therefore, we believe that the height difference seen in our STM topography truly reflects the relative difference of the c-axis lattice parameter between the $2\sqrt{2}\sqrt{2}$ CO and the $1\sqrt{2}\sqrt{2}$ phase, although it may be somewhat different from the real difference of the c-axis lattice parameter. Moreover, its magnitude is comparable to SCXRD results. To make our statement clearer, we added more datasets in Fig. S11 of the revised SM file, and added the above discussion in the revised main text, please see the last several lines in page 6 for more details.

Comment 3: It would be beneficial if the authors could provide some dI/dV curves of sample #2 and sample #1. Is there any feature of CDW gap in the long-ranged charge order phase?

Reply: This is a good suggestion. We have added the representative dI/dV spectra of the $2\sqrt{2}\sqrt{2}$ CO phase and the $1\sqrt{2}\sqrt{2}$ phase in both Fe₃Ge and Ge₂ layers of samples #1 and #2 in Figs. 2f,g of the revised main text. Independent of sample and the presence or not of CO, all the spectra collected in the Fe₃Ge layer are similar, exhibiting a partially opened gap-like feature at EF; all the spectra collected in the Ge₂ layer are similar, but showing a distinct peak at EF. We are not persuaded to assign the gap-like feature in Fe₃Ge layer as a CO gap for the following concerns. Firstly, such gap-like feature exists only on the Fe₃Ge layer but is absent on the Ge₂ layer which also shows the same CO. Secondly, it exists and shows the similar behavior in both the $2\sqrt{2}\sqrt{2}$ CO region and the $1\sqrt{2}\sqrt{2}$ region of the Fe₃Ge layer of samples #1 and #2, which seems likely that it has little correlation with the formation and the short- or long-ranged behavior of CO. Moreover, the latest ARPES and Raman scattering experiments did not see any signature of a CO gap in the long-ranged FeGe samples (arXiv: 2308.08336; arXiv: 2309.14314), consistent with our conclusion. We added the dI/dV spectra in Figs. 2f,g and related discussions in line 1 of page 5 of the revised main text.

Comment 4: While the long-ranged charge order in sample #2 appears inert to the defect density, its absence in the as-grown sample #1 with dense defect density need more detailed discussion. The authors should elaborate on how defect density affects long-ranged charge order.

Reply: We are sorry that these issues are not explained clearly in the last version. In the new version of our manuscript, we re-analyzed carefully the type and density of various defects and their effect on CO in samples #1 and #2 from the perspectives of STM, SCXRD and DFT calculations.

Firstly, judging from the crystallographic location of defects and their influence pattern on the surrounding local density of states (LDOS) distribution, we analyze STM topographic images and LDOS maps of multiple sample regions, and find mainly 7 types of defects on the Fe₃Ge layer and 2 types of defects on the Ge₂ layer. Figures S4i-q of SM display the detailed topographic images of the nine types of defects.

1) Although the specific appearance is different, types 1-5 of defects on Fe₃Ge layer are all C₆-symmetric with their defect centers located at the Ge1-site. These defects are considered as Ge1 vacancies, substitutional defect at Ge1a-site, occupational disorder at Ge1b-site and so on, but the specific correspondence is difficult to determine simply by STM study.

2) Type 6 of defects on Fe₃Ge layer is C₃-symmetric, and has two orientations, which is more likely

a Ge2-site defect when compared with Fig. S4c. It should be a Ge2 vacancy or a substitutional defect at Ge2-site. One thing should be mentioned is that such C3-symmetric defect is rarely observed on the Fe3Ge layer in our STM study.

3) Type 7 of defect on Fe3Ge layer is C2-symmetric and has three orientations, which is more likely an Fe-site defect, such as an Fe vacancy or a substitutional defect. Such type of defect is also rarely observed in our STM study and was only found in a small Fe3Ge region of sample #2.

4) Type 8 of defect is observed on the Ge2 layer, it is C6-symmetric with the defect center located at the Ge1-site, which might also be the Ge1-site defect seen on the Ge2 layer.

5) Type 9 of defect on the Ge2 layer is C3-symmetric and has two orientations, which should be the Ge2-site defect.

One thing should be mentioned is that the abovementioned defect types on Fe3Ge layer and Ge2 layer may overlap, such as types 1-5 and type 8, type 6 and type 9, since STM measurements may include information of both the topmost layer and the layer underneath.

Secondly, we count the density of these defects in samples #1 and #2. To simplify, we just count the total densities of Ge1-site, Ge2-site and Fe-site defects in multiple sample regions of samples #1 and #2, the results are shown in Fig. S5 of SM. The total density of Ge1-site defects is reduced by approximately a third to a half in sample #2, from ~ 1.7% of the atomic proportion of Ge1-site in sample #1 to ~ 0.8% in sample #2. Fe-site defects are rarely observed and will not be discussed in detail here. The density of Ge2-site defect is as low as less than ~ 0.01 % of the atomic proportion of Ge2-site and is almost unchanged in both samples, which should have little influence on CO.

Thirdly, scrutinizing the SCXRD data collected at 300 K carefully, we find that there is moderate residual electron density around Ge1b-site (corresponding to the Ge1 atoms that move along the c-axis away from the Kagome plane, the unchanged Ge1 atom is labeled as Ge1a-site, as illustrated in Fig. 5e), indicating obvious occupational disorders at Ge1-site. By adding such occupational disorders at Ge1-site, the refinements are moderately improved. The obtained proportion of Ge1b-site to the total Ge1-site and their averaged distance from the Kagome plane are ~ 3% and ~ 0.973 Å for sample #1 and ~ 2% and ~ 0.865 Å for sample #2, respectively. Such occupational disorders of Ge1-site have been reported by other SCXRD measurements (arXiv: 2308.09034). Other types of defects, such as Ge2-site defects, do not show up significantly in SCXRD data and the artificial introduction of a small amount of them has little effect on the refinements.

Finally, the SCXRD results agree qualitatively with our STM observations --- the main defects in FeGe are Ge1-site defects, which are significantly reduced in sample #2; while the density of Ge2-site defect is as low as less than ~ 0.01 % of the atomic proportion of Ge2-site and is almost unchanged in both samples, which has little influence on the refinement. Our study suggests that the reduction of Ge1-site disorder is beneficial to the development of a long-ranged CO in FeGe, which is also supported by the DFT calculations on Ge1-site defect, as displayed in Figs. 5f,g of the revised manuscript.

We added the new datasets and related discussions in paragraph 2 of page 5, the whole part of "Defect effects on CO" of the revised main text, parts 3 and 4 of the revised SM file, please see there for more details.

Comment 5: As shown in Fig.1a, the magnetic susceptibilities of different annealed samples show very different behaviors, such as the transition temperature of canted AFM changes a lot with the different annealed temperature, and an additional transition around 150 K in sample #2. Further discussion on these significant changes is encouraged.

Reply: We thank the reviewer for pointing out this important issue, which was also mentioned by the third reviewer. We agree that studying the variations of magnetic behavior and the relationship to CO is significant for further understanding the peculiar charge and magnetic properties in FeGe. However, as it is not the focus of this manuscript, and our experimental techniques (STM and SCXRD) are not sensitive to magnetism, we did not discuss more on that issue.

We notice that there are two papers that discussed the origin of the additional transition around 150 K in low-temperature annealed FeGe samples (arxiv:2308.01291, arxiv: 2308.09034). Such a transition always appears in samples annealed at low temperatures and is always accompanied by a transition around 260 K, and both transitions can be completely suppressed by a high enough magnetic field. It might be induced by the minor cubic B20-type FeGe byproduct probably formed from the low-temperature annealing process, as Cubic FeGe is expected to be transformed slowly from the hexagonal one for a low temperature annealing according to the phase diagram, and its magnetic susceptibility has a helimagnetic transition at around 275-280 K which roughly matches the value of 260 K here. This explanation seems reasonable, and we quoted the relevant papers and added brief discussion in line 4 from the bottom of page 3 of the revised main text.

As for the obvious influence of annealing temperature on the magnetic behavior below TCO, such as the canted AFM transition temperature mentioned by the reviewers, we don't know for sure the reasons at present. As above mentioned in the reply to comment 4, our STM and SCXRD measurements reveal that the defect density is strongly modified by annealing treatments, which may affect the delicate magnetic

interaction and thus change the low-temperature magnetic behavior. Besides, the formation of CO may also affect the magnetic behavior below TCO, and the variation of the proportion of CO will alter the magnetic susceptibility behavior. The contribution of these two effects to magnetism is difficult to draw conclusion from STM and SCXRD study, neutron scattering and other magnetic-sensitive techniques may provide decidable experimental evidence. As this issue is beyond the scope of the current manuscript, we just added a brief discussion in line 1 of paragraph 1 in page 4 of the revised main text.

Comment 6: The disappearance of canted AFM in sample #2 coincides with the observation of the long-ranged charge order. The authors should explore if the canted AFM have any influence on the disappearance of long-ranged charge order.

Reply: Thanks for pointing out this issue. We believe that the canted AFM is not the decisive factor in the formation of a long- or short-ranged CO.

Firstly, seen from the phase diagram of FeGe with a short-ranged CO studied by neutron scattering experiments (Nature 609, 490 (2022)), the CO once formed is not affected by spin canting and spin flop (induced by a high magnetic field). Moreover, we have performed STM and spin-polarized STM measurements on samples #1 and #2 at 4 K, and applied out-of-plane magnetic fields of $-14\text{T} \sim +14\text{T}$ to see the changes of spin direction and CO. We find that the spins flop from out-of-plane to in-plane direction under high magnetic fields, but the pattern and distribution of CO remain unchanged during this process (these data are being written in another paper). Our results are consistent with that of neutron scattering measurements, and suggest that the spin reorientation below TCO has little influence on the CO behavior.

Secondly, the canted AFM occurs below TCO, seeming more likely a result of the CO. One thing should be mentioned is that although the magnetic signal below TCO is strongly weakened in samples with a long-ranged CO, we can still see a kink at $\sim 20\text{-}60\text{ K}$ as indicated by the red arrows in Fig. R1. Whether it corresponds to the canted AFM or a new magnetic structure requires further neutron scattering study. Anyway, the low-temperature magnetic behavior seems closely related to the formation of long- or short-ranged CO, which may be affected by the formation of CO or the variation of defect density that alters the details of magnetic interaction. As discussed in the reply to comment 5, the delicate interplay between CO and canted AFM is beyond the scope of the current manuscript and needs further investigation by magnetic-sensitive techniques.

Fig. R1. Temperature dependent in-plane magnetic susceptibilities of as-grown FeGe (sample #1) and those after annealing at different temperatures as indicated. The CO transitions and the possible canted AFM transitions are indicated by the magenta and red arrows, respectively.

Replies to the second reviewer's report:

Comment 1: This paper reports careful transport, annealing, STM, X-ray scattering experiments to study the annealing effect on charge density wave (CDW) of FeGe, and establishment of long range CDW order in FeGe and lattice distortion induced by CDW. Over the past several years, CDW has been observed in several kagome lattice materials, and CDW in FeGe is particularly interesting because it appears at temperatures well below magnetic ordering temperature T_N . In the original paper, CDW in FeGe was reported to be short range order (refs. 30-34). The authors used different annealing temperature to show that low-temperature annealing can drive the system into long range CDW order, and solved lattice structure distortion induced by CDW order. Overall, the paper is well written and I think worthy of publication in Nature Communications. I do have some questions that authors should address before the paper is accepted for publication.

Reply: We thank the reviewer for pointing out the significance of our work and recommending publication in Nature Communications. The reviewer puts forward some insightful comments which help us improve our manuscript substantially, please see the point-by-point responses below.

Comment 2: The authors use STM to show that defect density is much lower in annealed sample with full CDW order. However, it is not clear at all what defects the authors are talking about? Are these defects Fe or Ge vacancies or excess Fe/Ge in the sample? This needs to be resolved to convince the reader about authors's story.

Reply: We thank the reviewer for pointing out this issue. Accordingly, we have carefully analyzed the types of defects, and their distribution in samples #1 and #2, the detailed results are discussed in part 3 of the revised SM file. The main results are also discussed below.

As STM lacks element resolution, one usually tries to judge the possible attribution of a defect from its crystallographic location and the influence pattern on the surrounding local density of states (LDOS) distribution. For FeGe, the possible patterns of the LDOS distribution influenced by the Ge1-site defect, Ge2-site defect and Fe-site defect are illustrated by in Figs. S4b-d of SM, from the perspective of crystallographic symmetry. It's obvious that the defect pattern should be C_6 -symmetric for Ge1-site defect,

C3-symmetric for Ge2-site defect, and C2-symmetric for Fe-site defect, respectively. Moreover, two degenerate C3-symmetric states should exist for Ge2-site defect, and three degenerate C2-symmetric states should exist for Fe-site defect, as illustrated in Figs. S4b-d.

Then we analyze STM topographic images and LDOS maps of multiple sample regions, and find mainly 7 types of defects on the Fe3Ge layer and 2 types of defects on the Ge2 layer. Figures S4i-q of SM display the detailed topographic images of the nine types of defects.

1) Although the specific appearance is different, types 1-5 of defects on Fe3Ge layer are all C6-symmetric with their defect centers located at the Ge1-site. These defects are considered as Ge1 vacancies, substitutional defect at Ge1a-site, occupational disorder at Ge1b-site and so on, but the specific correspondence is difficult to determine simply by STM study.

2) Type 6 of defects on Fe3Ge layer is C3-symmetric, and has two orientations, which is more likely a Ge2-site defect when compared with Fig. S4c. It should be a Ge2 vacancy or a substitutional defect at Ge2-site. One thing should be mentioned is that such C3-symmetric defect is rarely observed on the Fe3Ge layer in our STM study.

3) Type 7 of defect on Fe3Ge layer is C2-symmetric and has three orientations, which is more likely an Fe-site defect, such as an Fe vacancy or a substitutional defect. Such type of defect is also rarely observed in our STM study and was only found in a small Fe3Ge region of sample #2.

4) Type 8 of defect is observed on the Ge2 layer, it is C6-symmetric with the defect center located at the Ge1-site, which might also be the Ge1-site defect seen on the Ge2 layer.

5) Type 9 of defect on the Ge2 layer is C3-symmetric and has two orientations, which should be the Ge2-site defect.

One thing should be mentioned is that the abovementioned defect types on Fe3Ge layer and Ge2 layer may overlap, such as types 1-5 and type 8, type 6 and type 9, since STM measurements may include information of both the topmost layer and the layer underneath.

Furthermore, we count the density of these defects in samples #1 and #2. To simplify, we just count the total densities of Ge1-site, Ge2-site and Fe-site defects in multiple sample regions of samples #1 and #2, the results are shown in Fig. S5 of SM. The total density of Ge1-site defects is reduced by approximately a third to a half in sample #2, from ~ 1.7% of the atomic proportion of Ge1-site in sample #1 to ~ 0.8% in sample #2. Fe-site defects are rarely observed and will not be discussed in detail here. The density of Ge2-site defect is as low as less than ~ 0.01 % of the atomic proportion of Ge2-site and is almost unchanged in both samples, which should have little influence on CO. These results are also discussed in the revised main text, please see the last two paragraphs in page 7 for more details.

Comment 3: In the refinement obtained by authors, they claimed that the main effect of CDW order is c-axis dimerization, consistent with theory and previous X-ray work (Ref. 32). However, the refinements are done with full FeGe stoichiometry, how is this consistent with STM claim that annealed sample with full CDW would have less defects? The X-ray work seems to indicate that the sample stoichiometry is unchanged across CDW. Why would some samples show full CDW while other annealed samples show no CDW? How do one understand this from STM or X-ray scattering point of view? They seem to be inconsistent from the current story of the authors.

Reply: Thanks for pointing out this and we are sorry for the misunderstanding. Below is our detailed logic.

Firstly, as the defect density in samples #1 and #2 observed in STM study is less than 2% as discussed in the reply to comment 2, for simplicity, we first adopt the full FeGe stoichiometry to determine the structure of the CO phase. Reliable high-quality refinement results were obtained, as demonstrated by the small values of final R indexes (R1 and wR2 in Table S1 and S4 of SM). When considering the effect of defects, the refinements are improved moderately but the key structural features do not change, as shown in Fig. S9 of the revised SM.

Secondly, considering that the change of physical properties of FeGe by annealing is reversible, the FeGe stoichiometry will not change significantly during the annealing treatment processes. And because the RRR value of sample #2 is significantly increased and the density of defects observed by STM is reduced in sample #2 compared with sample #1, it is more likely that annealing treatments change mainly the occupational disorders of atomic sites.

Thirdly, scrutinizing the SCXRD data collected at 300 K carefully, we find that there is moderate residual electron density around Ge1b-site (corresponding to the Ge1 atoms that move along the c-axis away from the Kagome plane, the unchanged Ge1 atom is labeled as Ge1a-site, as illustrated in Fig. 5e), indicating obvious occupational disorders at Ge1-site. By adding such occupational disorders at Ge1-site, the refinements are moderately improved. The obtained proportion of Ge1b-site to the total Ge1-site and their averaged distance from the Kagome plane are ~ 3% and ~ 0.973 Å for sample #1 and ~ 2% and ~ 0.865 Å for sample #2, respectively. Such occupational disorders of Ge1-site have been reported by other SCXRD measurements (arXiv: 2308.09034). Other types of defects, such as Ge2-site defects, do not show up significantly in SCXRD data and the artificial introduction of a small amount of them has little effect on the refinements.

Finally, the SCXRD results agree qualitatively with our STM observations --- the main defects in FeGe are Ge1-site defects, which are significantly reduced in sample #2; while the density of Ge2-site defect is as low as less than ~ 0.01 % of the atomic proportion of Ge2-site and is almost unchanged in both samples, which has little influence on the refinement. Our study suggests that the reduction of Ge1-site disorder is beneficial to the development of a long-ranged CO in FeGe, which is also supported by the DFT calculations on Ge1-site defect, as displayed in Figs. 5f,g of the revised manuscript.

We added related discussions in paragraph 2 of page 5, the whole part of "Defect effects on CO" of the revised main text, parts 3 and 4 of the revised SM file.

Comment 4: I am particularly interested in knowing what the differences in crystal structures of FeGe for samples with short and long-range CDW order are. From STM measurements, the authors suggest as grown samples have considerable defects in Ge2 sites, how would that affect the refinements and how to understand the effects of these defects to CDW order?

Reply: As discussed in the reply to comments 2 and 3, our SCXRD study find that the main difference in FeGe samples with a short- and long-ranged CO is the proportion and distortion magnitude of Ge1b-site. The obtained proportion of Ge1b-site to the total Ge1-site and their averaged distance from the Kagome plane are $\sim 3\%$ and ~ 0.973 Å for sample #1 and $\sim 2\%$ and ~ 0.865 Å for sample #2, respectively. This agrees qualitatively with our STM observations --- the main defects in FeGe are Ge1-site defects, which are significantly reduced in sample #2. As the main effect of CO is the c-axis dimerization of Ge1 atoms, the disorder in Ge1-site will naturally affect the long-ranged order of Ge1-dimerization, which is confirmed by our DFT calculations shown in Figs. 5f,g.

In the previous version of our manuscript, the Ge1-site defects observed in the Ge2 layer were misinterpreted as the Ge2-site defects due to the poor spatial resolution, we are sorry for this misunderstanding. With higher spatial resolution, we find that the density of Ge2-site defect is as low as less than ~ 0.01 % of the atomic proportion of Ge2-site and is almost unchanged in both samples. This is consistent with the results of SCXRD that the introduction of Ge2-site defects has no significant effect on the refinement. Such a low density of Ge2-site defects should have little influence on the CO behavior.

We added the new datasets and related discussions in the part of "Defect effects on CO" of the revised main text and parts 3 and 4 of the revised SM file.

Comment 5: The DFT calculations suggest defect 2 has lower energies and that prevent the formation of $\frac{1}{4}$ Ge1 dimerization and CDW order. While this is a nice picture, I don't understand how this can be consistent with Fig. 4 where STM current can perturb the CDW order? I would assume STM tip current would not be able to affect defects in FeGe. There should be a detailed discuss to make sure the story is self-consistent between different probes.

Reply: As pointed out in this manuscript and other related papers (arxiv:2307.10565, Phys. Rev. Materials 7, 104006 (2023)), the original $1 \times 1 \times 1$ phase and the $2 \times 2 \times 2$ CO are nearly degenerate in energy, moderate disturbances will upset their balance and cause the transition from $2 \times 2 \times 2$ CO to the $1 \times 1 \times 1$ phase. Figures 4 and 5 correspond to two methods to tune the ground state. The disturbance might be the inherent defect in the sample, such as Ge1-site defect as indicated in Fig. 5e-g; or it could be external interference, such as the electric field between STM tip and sample (Nat Commun 13, 1843) or electron/hole injections induced by STM tip (Nat Commun 7, 10956 (2016)), etc. Figure 4 illustrates the latter case, where applying a high bias voltage between the STM tip and FeGe sample will induce a large electric field or electron/hole injections, but not to affect the defects. We are sorry for this misunderstanding, and we added more discussion to make it clearer in the revised manuscript, please see line 5 of the last paragraph in page 6 for more details.

Comment 6: Are there signatures of Ge2 defect concentration from X-ray data? Are they consistent with STM work?

Reply: As discussed in the reply to comments 2-4, by STM measurements with higher spatial resolution, we can resolve Ge1-site and Ge2-site defects observed in Ge2 layer, as discussed in part 3 of the revised SM file. The real density of Ge2-site defect is very low in both samples #1 and #2, and the introduction of Ge2-site defects has no significant effect on the refinement of SCXRD data. In the revised manuscript, we modified related discussion.

Comment 7: I noticed that a recent preprint claiming a lattice distortion in FeGe (<https://arxiv.org/abs/2309.14314>). Is it consistent with authors X-ray work?

Reply: We have also noticed the preprint paper (arXiv: 2309.14314) mentioned by the reviewer. The paper reports the anomalous evolution of the crystal symmetry of FeGe with temperature by utilizing neutron Larmor diffraction and Raman spectroscopy, which is considered to be related to the delicate coupling of multiple lattice, charge and spin degrees of freedom. They measured mainly the weak in-plane lattice distortions on the order of 10^{-4} , which is far exceeding the resolution of STM and SCXRD techniques and

was not captured by our measurements. Our study mainly focused on the c-axis lattice distortion, and the obtained CO structure was cited by this paper. Therefore, we think that both experimental results are consistent on the large lattice distortions, but it is hard to compare directly the very small in-plane lattice distortions.

Comment 8: I also noticed a competing paper (<https://arxiv.org/abs/2308.09034>) posted on arxiv. The authors should make some comments near the end about their work and the competing work, so a general audience would know what is going on in the field.

Reply: We thank the reviewer for pointing out this issue. We have added notes at the end of the revised manuscript to compare our work with this work.

Comment 9: Overall, the experimental part of the paper is nice and has considerable new information. However, it is not clear to me how different probes are self-consistent. The paper is not suitable for publication as written. A revised version of the paper might be suitable for publication.

Reply: We thank the reviewer for giving high regard of our work. We have done more analysis to prove that different probes in our study are self-consistent. We believe our additional datasets and improved analysis/discussion can eliminate the ambiguity in the previous version, and we hope the reviewer will find it satisfying.

Replies to the third reviewer's report:

Comment 1: The authors present a study that reveals a way to tune the recently discovered charge order in the magnetic kagome system FeGe into a long range order by thermal annealing. Understanding emergent orders, especially charge order, in kagome lattices is an important current topic in condensed matter physics. The CO in FeGe is especially interesting as it is manifested in a moderately correlated system. The authors present a systematic comparison of an as-grown FeGe and a thermally annealed FeGe. First, they show that the annealing process reduces the density of defects via an STM study, which is responsible for the tuning of the CO. Second, they show that the CO exhibits a Ge dimerization as previously proposed. They do this by purposely destroying a region of CO using the STM tip, and then examining the height differences. Third, as defects exist in both the Fe₃Ge layer and the Ge₂ layer, they also explored which defects are more important for the CO stability, and made the observation that the defects in the Ge₂ layer is more important, and support this with DFT calculations. Overall, this is a very important and systematic study. It provides key information to support the importance of Ge dimerization to the CO in FeGe, and also provides insights into why thermal annealing can be used to tune CO in this system. I find the analyses thorough and solid. I would recommend publication of this work. I only have the following minor questions below.

Reply: We thank the reviewer for the comprehensive summary of the key results of our manuscript and recommending publication in Nature Communications. Below we respond to the reviewer's questions point-by-point.

Comment 2: What are the native defect states shown in Fig. 1 e-f? And what are the residual atoms as the bright spots? Do the authors have any idea if they are Fe or Ge? It is not clear to me why cleaving would leave such residual atoms behind. Can STS taken on these defects provide any insights into what these two types of defects are?

Reply: We thank the reviewer for pointing out these issues.

Firstly, we carefully studied the attribution and density of different types of defects, the detailed results are discussed in part 3 of the revised SM file. The main results are also summarized here. By analyzing STM topographic images and LDOS maps of multiple sample regions, we find mainly 7 types of defects on the Fe₃Ge layer and 2 types of defects on the Ge₂ layer. Figures S4i-q of SM display the detailed topographic images of the nine types of defects.

1) Although the specific appearance is different, types 1-5 of defects on Fe₃Ge layer are all C₆-symmetric with their defect centers located at the Ge₁-site. These defects are considered as Ge₁ vacancies, substitutional defect at Ge_{1a}-site, occupational disorder at Ge_{1b}-site and so on, but the specific correspondence is difficult to determine simply by STM study.

2) Type 6 of defects on Fe₃Ge layer is C₃-symmetric, and has two orientations, which is more likely a Ge₂-site defect when compared with Fig. S4c. It should be a Ge₂ vacancy or a substitutional defect at Ge₂-site. One thing should be mentioned is that such C₃-symmetric defect is rarely observed on the Fe₃Ge layer in our STM study.

3) Type 7 of defect on Fe₃Ge layer is C₂-symmetric and has three orientations, which is more likely an Fe-site defect, such as an Fe vacancy or a substitutional defect. Such type of defect is also rarely observed in our STM study and was only found in a small Fe₃Ge region of sample #2.

4) Type 8 of defect is observed on the Ge₂ layer, it is C₆-symmetric with the defect center located at the Ge₁-site, which might also be the Ge₁-site defect seen on the Ge₂ layer.

5) Type 9 of defect on the Ge₂ layer is C₃-symmetric and has two orientations, which should be the Ge₂-site defect.

One thing should be mentioned is that the abovementioned defect types on Fe₃Ge layer and Ge₂ layer

may overlap, such as types 1-5 and type 8, type 6 and type 9, since STM measurements may include information of both the topmost layer and the layer underneath.

Secondly, we discuss the origin of residual atoms. As we know, the cleavage surface usually occurs along the direction of the weakest internal connection of the crystal, whether a large clean cleavage surface can be obtained depends on the strength of the interlayer bonding. Judging from the strong k_z dispersion of the band structure and the rather three-dimensional (3D) crystal shape, FeGe is a relatively 3D system with strong interlayer bonding between the Fe₃Ge and Ge₂ layers where the cleavage happens. During cleavage, the existence of local uneven forces, crystal defects, or temperature effect, may result in unsmooth surface, residual atomic/molecular organization, or surface reconstruction, etc. This is commonly observed in 3D or evenly quasi-2D materials, such as residual atoms and multiple surface reconstructions in iron-based 122 systems (Phys. Rev. Lett. 103, 076104 (2009), arXiv:1609.00846), residual alkali metal atoms and multiple surface reconstructions in AV₃Sb₅ (Nano Lett. 22, 918 (2022)), residual atoms in Kagome Fe₃Sn₂ (Nature 562, 91 (2018)) and ScV₆Sn₆ (npj Quantum Mater. 9, 14 (2024)), etc.

For the residual bright spots on the cleavage surfaces of FeGe, we judge their origin by the crystal structure and dI/dV spectra. As shown in Fig. R2, we find a small Fe₃Ge terrace near the atomic step of Ge₂ layer, surrounded with some standalone islands (indicated by the white arrows) of basically the same height. On these islands, the dI/dV spectra show a dip-like feature near EF (spectra 1 and 2 in Fig. R2e) as that observed in Fe₃Ge layer (Fig. 2f), supporting that these small islands are probably residual Fe₃Ge fragments. For the very small bright spots shown in Figs. R2f,g, we cannot distinguish their atomic structure and try to determine their origin by dI/dV spectra. We find that some spots have spectra that are similar to that of the Fe₃Ge layer (spectrum 4 in Fig. R2h), and some resemble to that of the Ge₂ layer with a peak-like feature near EF (spectrum 3 in Fig. R2h). Therefore, we conclude that most of the residual bright spots are small Fe₃Ge fragments/clusters, while the others may be Fe, Ge, or Fe-Ge clusters.

[Image redacted]

Fig. R2 | Topographic image and dI/dV spectra collected on bright spots. a, Typical topographic image of a sample region in Ge₂ layer with small exposed Fe₃Ge regions. b, Line profile taken along cut #1 in panel a. The regions of Fe₃Ge layer and Ge₂ layer are marked out. c,d, Typical topographic images of bright spots near the exposed Fe₃Ge layer. e, Typical dI/dV spectra taken on the flat Ge₂ and Fe₃Ge layer, as well as collected on the bright spots as labeled by purple and orange dots in panels c and d. The spectra are shifted vertically for clarity. f,g, Typical topographic images of bright spots away from the exposed Fe₃Ge layer. h, Typical dI/dV spectra taken on Ge₂ and Fe₃Ge layer, as well as the bright spots as labeled by the magenta and coffee dots in panels f and g.

Comment 3: From the susceptibility plot in Fig. 1 a, it appears that the annealing conditions also strongly affect the feature well below 100K. There is a big peak around ~40K (?) in the as-grown crystal, which is basically preserved in the 560C sample, but gone in the 320C and strongly suppressed in the 400C sample. Do the authors have any understanding of what this is and how it is affected as well by the annealing process?

Reply: We thank the reviewer for pointing out this important issue, which was also mentioned by the first reviewer. As shown in Fig. R1, strong magnetic signal and a big peak below TCO appears in the as-grown and 560 Å annealed samples, which was considered related to the canted AFM transition. While in FeGe samples annealed at 320 Å and 400 Å with a long-ranged CO, we find that although the magnetic signal below TCO is strongly weakened, a kink at ~ 20-60 K is still observed as indicated by the red arrows in Fig. R1. Whether it corresponds to the canted AFM or a new magnetic structure requires further neutron scattering study. The low-temperature magnetic behavior seems closely related to the formation of long- or short-ranged CO, which may be affected by the formation of new CO superstructure or the variation of defect density (as revealed by STM and SCXRD) that alters the details of magnetic interaction. Since the delicate interplay between CO and canted AFM is beyond the scope of the current manuscript and needs further investigation by magnetic-sensitive techniques, we did not discuss more in the manuscript, but just added a brief discussion in line 1 of paragraph 1 in page 4 of the revised main text.

[Image redacted]

Fig. R1. Temperature dependent in-plane magnetic susceptibilities of as-grown FeGe (sample #1) and those after annealing at different temperatures as indicated. The CO transitions and the possible canted AFM transitions are indicated by the magenta and red arrows, respectively.

Version 1:

Reviewer comments:

Reviewer #1

(Remarks to the Author)

I would like to thank the authors for their answers to my concerns. The authors have addressed almost all the concerns in my first report. I have some minor comments on the newly added data:

Why the presence or absence of long-range charge order seems does not have a significant impact on the dI/dV spectra both on the Fe₃Ge layer and Ge₂ layer? What's the source of the peak around the Fermi level in the dI/dV spectra of the Ge₂ layer? Can the difference in dI/dV between the two cleavage planes be explained by DFT?

When those last things have been addressed, I recommend publishing this manuscript in Nature Communications.

Reviewer #2

(Remarks to the Author)

I have read carefully the revised paper and rebuttal letter to comments by referees. I think issues raised in the reviewing process are addressed and therefore I am happy to recommend the paper for publication.

Reviewer #3

(Remarks to the Author)

I have read carefully the authors' response to my comments and questions, revisions to the main text and supplementary materials, and am satisfied with the authors' response and efforts in identifying the types of disorders in FeGe. The paper is now acceptable to be published in Nat. Comm.

Author Rebuttal letter:

NCOMMS-23-55664-A

Discovery of a long-ranged charge order with 1/4 Ge₁-dimerization in an antiferromagnetic Kagome metal by Ziyuan Chen et al.

Reply to the reviewers:

We thank all reviewers for their time and insightful comments on our manuscript. Our point-by-point responses are in blue text below and the original comments are in italic. The corresponding revisions in manuscript are highlighted with yellow background.

Replies to the first reviewer's report:

Comment 1: I would like to thank the authors for their answers to my concerns. The authors have addressed almost all the concerns in my first report.

Reply: We thank the reviewer for reading carefully the revised manuscript and responses to comments, and we are happy that the reviewer is satisfied with our reply. Below we respond the reviewer's comments point-by-point, and we believe our reply can eliminate the reviewer's doubts.

Comment 2: I have some minor comments on the newly added data: Why the presence or absence of long-range charge order seems does not have a significant impact on the dI/dV spectra both on the Fe₃Ge layer and Ge₂ layer?

Reply: We thank the reviewer for pointing out this issue. This is a good question that does make us confused for a long time. As our own research progresses and more other results are reported, we now have a reasonable understanding of this phenomenon. A recent ARPES work (arXiv: 2404.02231) has measured the detailed band structure of various FeGe samples with or without a long-range CO, which were also obtained by different annealing treatments. They find that the electronic structures are very similar in different FeGe samples with/without a long-ranged CO and below/above the CO transition, the presence of a long-ranged CO perturbs slightly the electronic structure around EF. This result has also been confirmed by other ARPES reports and several DFT calculations (arXiv: 2308.08336; arXiv:2307.10565). Moreover, we have recently measured the band structures of FeGe samples with/without a long-ranged CO by quasiparticle interference (QPI) in STM, and find that the scattering patterns of the two samples near EF are very similar, revealing a similar band structure (the related datasets and discussions will be included in a separate work which is under preparation). Therefore, it is very natural that the dI/dV spectra both on the Fe₃Ge layer and Ge₂ layer should not be significantly influenced by the presence or absence of a long-ranged CO, due to the slightly changed band structure. Previous studies have also reported some subtle changes of band structure at certain momenta for the CO phase, however, as dI/dV spectrum measures the integrals of the whole momentum space, it is a little difficult for STM to distinguish these subtle changes.

Comment 3: What's the source of the peak around the Fermi level in the dI/dV spectra of the Ge₂ layer?

Can the difference in dI/dV between the two cleavage planes be explained by DFT?

Reply: We thank the reviewer for pointing out this issue. Since STM is a surface sensitive tool and different cleavage planes may hold different orbital compositions that form different bands, the obtained dI/dV spectra can be different on different cleavage planes. This is commonly found in STM experiments, such as Kagome metals ScV₆Sn₆ (Appl. Surf. Sci. 665, 160190 (2024)), CoSn (Nat. Commun. 11, 4003 (2020)), Co₃Sn₂S₂ (Phys. Rev. B 99, 245158 (2019)), etc.

As for the peak around EF in the dI/dV spectra of Ge₂ layer, it does not split after applying a vertical magnetic field of 14 T, thus the possibility of a Kondo resonance peak is precluded. We suspect that it arises from some features in the band structure. To confirm this and to further understand the different dI/dV spectra in Fe₃Ge layer and Ge₂ layer, we try to compare those with the results of DFT calculations. Figure R1 shows the integral density of states (DOS) of the entire BZ for the band components of Fe, Ge₁ and Ge₂ atoms. The integral DOSs of Ge₁ and Ge₂ components exhibit a distinct peak near EF; while for the Fe component, the integral DOS shows many peaks between ± 0.2 eV, one locates at EF but its intensity differs slightly with the background. It seems that the measured dI/dV spectra on Ge₂ layer are somewhat similar to those of Ge bands in DFT calculations, and the peak near EF may arise from the large DOSs of Ge bands at certain momenta, such as relatively flat bands or band tops/bottoms. However, we also notice that there are significant differences between the dI/dV spectra and DFT results, which may be caused by two reasons. Firstly, there are some differences between the band structures calculated by DFT and measured by ARPES, the reason is unclear now and is still under investigation. Secondly, STM measurements usually have some orbital and momentum selectivity. Both reasons will lead to the difference between DFT calculations and dI/dV spectra in STM measurements. Further experimental and theoretical studies are needed to further understand the characteristics of the dI/dV spectra and the influence of CO on electronic structure. Now we are analyzing the QPI data to try to understand these phenomena and to reveal the delicate difference induced by CO; however, due to the complex band structures, in-depth analysis is still in progress and the results will be present in a separate work. Because of the above reasons and as this is not the focus of this manuscript, we do not intend to explain more in this manuscript, and we hope that the reviewer can understand.

[Image redacted]

Fig. R1. Integral DOS of the entire BZ for the band components of Fe, Ge₁ and Ge₂ atoms, based on the band structure by DFT calculations.

Comment 4: When those last things have been addressed, I recommend publishing this manuscript in Nature Communications.

Reply: We thank the reviewer and hope our response satisfactory.
Replies to the second reviewer's report:

Comment 1: I have read carefully the revised paper and rebuttal letter to comments by referees. I think issues raised in the reviewing process are addressed and therefore I am happy to recommend the paper for publication.

Reply: We thank the reviewer for their time and recommending our paper for publication in Nature Communications.

Replies to the third reviewer's report:

Comment 1: I have read carefully the authors' response to my comments and questions, revisions to the main text and supplementary materials, and am satisfied with the authors' response and efforts in identifying the types of disorders in FeGe. The paper is now acceptable to be published in Nat. Comm.

Reply: We thank the reviewer for their time and recommending our paper for publication in Nature Communications.

Version 2:

Reviewer comments:

Reviewer #1

(Remarks to the Author)

The authors have addressed all my questions. I am happy to recommend the publication of the current manuscript.
